# ProtocolLLM: RTL Benchmark for SystemVerilog Code Generation of Communication Protocols

Arnav Sheth[*], Ivaxi Sheth[†], Mario Fritz[†],

[*]University of Illinois Urbana Champaign. *amsheth2@illinois.edu*

[†]CISPA Helmholtz Center for Information Security. *ivaxi.sheth@cispa.de,fritz@cispa.de*

*Abstract*—Recent advances in Large Language Models (LLMs) have shown promising capabilities in generating code for general-purpose programming languages. In contrast, their applicability for hardware description languages, particularly for generating synthesizable and functionally correct designs, remains significantly underexplored. HDLs such as SystemVerilog are logic-oriented and demand strict adherence to timing semantics, concurrency, and synthesizability constraints. Moreover, HDL-based design flows encompass a broad set of tasks beyond structural code generation, including testbench development, assertion-based verification, timing closure, and protocol-level integration for on-chip communication. The objective of our paper is to analyze the capabilities of state-of-the-art LLMs in generating SystemVerilog implementations of standard communication protocols, a core component of embedded and System-on-Chip (SoC) architectures. This paper introduces the first benchmark suite targeting four widely used protocols: SPI, I²C, UART, and AXI. We define code generation tasks that capture varying levels of design abstraction and prompt specificity. The generated designs are assessed for syntactic correctness, synthesizability, and functional fidelity via waveform simulation and test benches[1].

## I. INTRODUCTION

Communication protocols are essential to hardware-embedded systems, enabling structured data exchange between processing elements, peripherals, and memory subsystems. In embedded and system-on-chip (SoC) designs, standard protocols such as Serial Peripheral Interface (SPI) [18], Inter-Integrated Circuit (I²C) [26], Universal Asynchronous Receiver-Transmitter (UART) [12], and Advanced extensible Interface (AXI) [1] are widely used to interface with sensors, actuators, storage devices, and external. The correct and efficient implementation of these protocols in hardware description languages (HDLs) is essential for ensuring functional correctness, timing closure, and system-level integration.

LLMs have shown remarkable performance in code generation, particularly in software domains [10], [11]. As the complexity of hardware designs continues to grow, there is increasing interest in leveraging language models to assist or automate various stages of the design process [3], [4], [7], [16], [17], [20], [23]–[25]. Their ability to synthesize, complete, and refactor code raises the question: *Can such models generate synthesizable HDL code that adheres to the semantic, structural, and timing constraints of hardware communication protocols?*

Despite the emergence of LLM-based tools for software engineering, their application to hardware domains, especially

in generating protocol-level modules, remains limited. During hardware development, communication protocols such as SPI, I²C, UART, and AXI serve as fundamental interfaces for inter-module data exchange. These protocols are widely used across various systems, from low-power microcontrollers to high-performance SoCs. Designing synthesizable RTL for these protocols is a non-trivial task that demands strict adherence to timing specifications, finite-state machines (FSMs), signal coordination, and electrical constraints such as clock polarity and phase.

The communication protocol modules are one of the most basic and common examples in VLSI design that also require more holistic evaluation, including waveform-level functional verification, synthesis timing analysis, and deployability on hardware targets such as FPGAs. In particular, protocol designs involve multi-signal interactions, such as Ready/Busy/ACK Signals, MOSI/MISO, and Multibyte data transmission, which must adhere to strict temporal relationships, making surface-level correctness insufficient for meaningful evaluation.

Current HDL code generation works have largely focused on isolated or synthetic examples in Verilog and VHDL [6], [14], [15], [25], [27], [28]. Many of them focus on syntax correctness alone [3], [4], [16] for evaluation. In contrast, protocol-level designs such as SPI, I²C, or AXI demand adherence to precise timing relationships, signal-level interactions, and behavioral specifications derived directly from detailed datasheets. These implementations require waveform-accurate behavior and rigorous validation against temporal constraints, which are not captured by syntax-based benchmarks.

Given the adoption of SystemVerilog across the semiconductor industry for both hardware design and verification, our task requires code generation specifically in SystemVerilog. Unlike traditional Verilog benchmarks, SystemVerilog's advanced constructs directly facilitate the modeling of complex protocols and interactions.

We introduce the first benchmark in System Verilog for evaluating LLMs on HDL-based communication protocol generation to address this gap. Our benchmark covers multiple protocols: SPI, I²C, UART, and AXI. The benchmark focuses on open-ended code generation, where models are required to synthesize complete, synthesizable modules that meet protocol-level functional and temporal constraints.

In our open-source benchmark, we perform extensive experiments across different LLMs, including code models and

---

[1]Code: https://github.com/amsheth/FPGA_protocols_llm

general-purpose models. The code is comprehensively evaluated in three stages: (1) Lint Pass to ensure language-level correctness and synthesizability, (2) Logic Synthesis using commercial-grade EDA tools, and (3) Waveform Analysis to validate temporal behavior against golden references. Finally, our contributions can be summarised below:

1) We propose the **first** structured benchmark focused on communication protocol generation using LLMs, spanning widely used industry protocols: SPI, I²C, UART, and AXI.
2) We introduce a three-stage evaluation framework tailored to hardware development workflows. This includes (i) lint checks, (ii) logic synthesis, and (iii) waveform validation for assessing hardware resource usage, maximum achievable frequency, and area overhead.
3) We evaluate a diverse set of LLMs, including both code-focused models and general-purpose models under both vanilla and retrieval-augmented generation (RAG) setups.

## II. BACKGROUND

### A. HDL Code Generation with LLM

Recent works have revealed weaknesses of LLMs for HDL generation, particularly in synthesis compatibility, functional correctness, and verification logic quality. Benchmarks like RTLLM [16] and VerilogEval [15] show that LLMs frequently generate non-synthesizable HDL due to incorrect timing constructs, vendor-incompatible syntax, and structural flaws [3], [4], [7], [16], [23], [24]. Formal tools such as Cadence Jasper have further identified that up to $60\%$ of LLM-generated designs contain critical weaknesses like incorrect state transitions and bit-width mismatches [5], [9], while iterative techniques such as counterexample-guided refinement offer only partial mitigation [7], [9]. Despite syntax improvements, logical correctness remains a persistent issue, with misconfigured ports, flawed carry logic, and incomplete FSM implementations reported by previous works. Verification logic, including testbenches and SystemVerilog Assertions (SVAs), often lacks coverage and adherence to industry practices like UVM [21], highlighting insufficient feedback integration.

Unlike prior benchmarks that focus on isolated RTL modules or general-purpose logic generation, our work targets protocol-specific HDL generation, incorporating temporal and functional constraints derived from real-world datasheets. Furthermore, we introduce a multi-stage evaluation pipeline, including waveform-level validation, which more closely aligns with practical hardware verification flows.

### B. Communication Protocol

A common and critical application of FPGAs and microcontrollers is the implementation and management of communication protocols. These protocols are essential for interfacing components such as sensors [13], actuators [2], and master controllers in embedded [22]. In this benchmark, we focus on four widely used inter- and intra-system communication protocols: SPI, I²C, UART, and AXI:

- **SPI** [18] is a synchronous serial communication interface that employs a master-slave architecture, where a single master device orchestrates communication with one or more slave devices. The master device generates the clock signal on the SCLK line, which synchronizes data transmission between the master and the selected slave. Unlike I2C's addressing scheme, SPI often uses a dedicated Slave Select (SS) line for each slave device. The master activates a specific slave by pulling its SS line low before initiating data transfer. Data is simultaneously transmitted and received between the master and the slave on the MOSI and MISO lines, respectively, during each clock cycle.
- **I2C** [26] is a synchronous, multi-master/multi-slave serial communication bus that utilizes only two bidirectional open-drain lines: the Serial Data Line (SDA) and the Serial Clock Line (SCL). I2C employs a master-slave architecture where one or more master devices initiate and control communication with multiple slave devices. The master device is responsible for generating the clock signal on the SCL line that synchronizes all data transfers. Each slave device connected to the I2C bus is assigned a unique address, which the master uses to select and communicate with a specific slave. This addressing scheme allows multiple slave devices to share the same two communication wires, a significant advantage over SPI, which typically requires a dedicated select line for each slave.
- **UART** [12] is an asynchronous, full-duplex serial communication protocol commonly employed for point-to-point data exchange. Unlike synchronous protocols, UART does not require a shared clock signal instead, its baud rate must be configured identically on both communicating devices. Data is transmitted serially, one bit at a time, over two dedicated lines: the Transmit (Tx) line and the Receive (Rx) line. Communication is packet-based, with each data frame consisting of a start bit, a configurable number of data bits (typically 8), an optional parity bit for error detection, and one or more stop bits.
- **AXI4 (Lite)** [1] is an interface protocol defined by ARM as part of the AMBA (Advanced Microcontroller Bus Architecture) standard. While AXI is inherently a point-to-point interface, its architecture can be extended to support multiple masters and slaves through the use of interconnect components, which act as intelligent switches and routers on the chip. The protocol mandates a handshake-like procedure involving VALID and READY signals for each transmission, ensuring reliable data transfer by synchronizing the data flow between the source and destination. AXI-Lite bus is an AXI bus that only supports a single ID thread per initiator.

These protocols were selected based on their practical relevance and increasing complexity, enabling a gradient of difficulty for evaluating the ability of LLM to synthesize code.

## III. PROTOCOLLLM

In this section, we present a comprehensive benchmark designed to evaluate LLMs for the task of generating synthesizable HDL implementations of communication protocols. These protocols serve as foundational components in modern hardware systems, facilitating the communication between processing units, peripherals, and memory subsystems. The complexity of accurately implementing these protocols in HDL, particularly as SystemVerilog, requires strict adherence to timing constraints, signal coordination, and FSM behavior, making it an ideal domain for testing LLM-based code generation.

Our benchmark is designed to evaluate both the functional and timing correctness of LLM-generated designs across these protocols.

### A. Design Principles

The design of our benchmark is guided by the following core principles:

- **Protocol-Centric Evaluation**: The benchmark focuses on a set of widely-used communication protocols: SPI, I²C, UART, and AXI that are essential for interfacing between processing elements and peripherals. These protocols represent different levels of complexity in terms of signaling, timing, and interaction, providing a diverse set of challenges for model evaluation.
- **Synthesizability and Timing Fidelity**: Beyond syntax, the generated HDL must conform to hardware implementation constraints, including FSM integrity, signal synchronization, and timing closure. This ensures that models produce RTL code that is not only valid but also deployable on real-world FPGA hardware.
- **Specification-Guided Generation**: Instead of synthetic or abstract prompts, our tasks are grounded in protocol specification documents and datasheets. This setting reflects practical design conditions where engineers synthesize HDL directly from interface specifications, enabling a more realistic assessment of model utility.
- **Holistic Evaluation Pipeline**: Our benchmark includes a three-stage evaluation framework encompassing (i) linting for syntax and synthesis readiness, (ii) logic synthesis for area, frequency, and utilization metrics, and (iii) waveform analysis for functional validation against protocol timing diagrams and expected behaviors.

### B. Benchmark Tasks and Evaluation Criteria

Our primary task is Open-Ended full module generation, where the model receives a natural language or formal protocol description and is tasked with generating a complete, synthesizable SystemVerilog module. This evaluates the model's ability to interpret high-level requirements and produce corresponding RTL code.

We prompt the model in two ways:

- **Standard Prompting**: The model generates HDL code based on a full problem description, often in natural language or mixed formats.

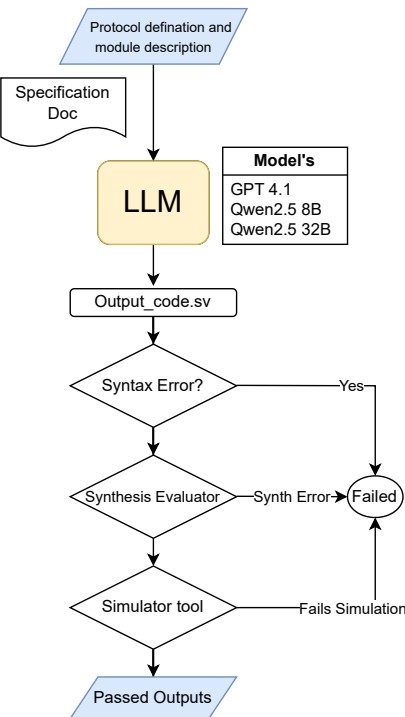

Fig. 1. Our HDL code generation and evaluation methodology. The pipeline starts with protocol definitions and optional specification documents as input to an LLM (GPT-4.1, Qwen2.5 14B, or Qwen2.5 32B). The generated SystemVerilog code is evaluated through three sequential stages: (1) syntax checking, (2) logic synthesis, and (3) functional simulation. Any failure at these stages leads to rejection, while only code that passes all three is accepted as valid output.

- **Spec-Assisted Generation**: The model accesses an external corpus of reference designs to simulate real-world reuse, testing its ability to retrieve and apply relevant knowledge for optimized designs.

**Prompts**

Generate a SPI driver in system verilog with the following structure, and given that we want its parameters as CPOL=1 and CPHA=1. This SPI driver should be able to act as the master.

**Module Defination**

```
module SPI_driver(
    input logic clk, rst,
    input logic [7:0] data_in,
    input logic SPI_MISO,
    Input logic SPI_start,
    output logic SPI_MOSI,
    output logic SPI_CLK,
    output logic SPI_EN,
    output logic [7:0] data_out );
```

Fig. 2. Sample prompts for SPI

Evaluation of a code generated by LLM is challenging, as LLMs are usually trained to write code for object-oriented programming languages like C/C++ and Python, which are usually not time-dependent and do not generate combinational and sequential logic. Our criteria to evaluate SystemVerilog code are on 3 three-prong basis to completely evaluate and

TABLE I
EVALUATION RESULTS FOR HDL COMMUNICATION PROTOCOL GENERATION. SYMBOLS: ✓= PASS, ✗= FAIL WHEN PROMPTED WITH AND WITHOUT SPECIFICATION FILES.

| Model | Protocol | Lint Pass | | Synthesis | | Waveform | |
|---|---|---|---|---|---|---|---|
| | | w/o Spec | w Spec | w/o Spec | w Spec | w/o Spec | w Spec |
| QwenCoder 2.5-14B | SPI | ✗ | ✓ (Warn) | ✗ | ✓ | ✗ | ✗ |
| | I²C | ✗ | ✗ | ✗ | ✗ | ✗ | ✗ |
| | UART | ✗ | ✗ | ✗ | ✗ | ✗ | ✗ |
| | AXI | ✗ | ✗ | ✗ | ✗ | ✗ | ✗ |
| QwenCoder 2.5- 32B | SPI | ✗ | ✗ | ✗ | ✗ | ✗ | ✗ |
| | I²C | ✗ | ✓ (Warn) | ✗ | ✗ | ✗ | ✗ |
| | UART | ✗ | ✓ (Warn) | ✗ | ✗ | ✗ | ✗ |
| | AXI | ✗ | ✓ (Warn) | ✗ | ✓ | ✗ | ✗ |
| GPT-4.1 | SPI | ✓ | ✓ | ✓ | ✓ | ✓ | ✓ |
| | I²C | ✓ (Warn) | ✗ | ✓ | ✗ | ✓ | ✗ |
| | UART | ✗ | ✓ | ✗ | ✓ (Warn) | ✗ | ✗ |
| | AXI | ✓ (Warn) | ✗ | ✓ | ✗ | ✗ | ✗ |

accept the generated code as working. The criteria in the order are as follows:

- **Lint Check**: We use the industry standard Synopsys Spyglass Fault Analyzer. This is the first step, which looks for any syntactical or logical errors generated by the code.
- **Synthesis**: Synthesis analysis is done on the Synopsys Design Compiler to know about the resource consumption and if the code can be converted into a bitstream and offloaded onto an FPGA.
- **Waveform Analysis**: The waveform analysis will determine whether the generated code correctly implements the communication protocol. We will simulate the generated modules on testbenches the randomly generate vectors and signals and check if the module has correctly sent or received the data in accordance to the specification

### C. Experimental Setup

We evaluate the benchmark on open-source coder models: Qwen Coder 2.5-14B and Qwen Coder 2.5-32B [8]. We also evaluate on a general-purpose proprietary GPT-4.1 model [19]. All models are queried with a temperature of 0 to ensure deterministic outputs for reproducibility.

## IV. RESULTS

Our benchmark evaluation reveals several important patterns about the capabilities and limitations of current LLMs in HDL protocol generation, particularly when targeting synthesizable and functionally correct communication modules. At a system-wide level, GPT-4.1 outperforms both Qwen-Coder models across all evaluation axes: syntax, synthesis, and waveform fidelity. It is the only model capable of generating end-to-end functional designs. Qwen-Coder 32B, while structurally better, still falls short of generating timing-correct or resource-safe HDL.

Interestingly, we observe a clear protocol-specific performance bias: simpler serial protocols like SPI and I²C were more reliably generated, while UART and AXI, which involve greater temporal complexity and signal interplay, exposed deeper model weaknesses. Semantic conditioning via spec files had nuanced effects. For simpler protocols like SPI, spec guidance significantly improved synthesis success, e.g., QwenCoder2.5-14B went from full fail to passing synthesis.

As a side note, what we consider warnings are basically what a good and ideal code should not contain, but are both syntactically correct and do not affect its ability. For example, a signed to unsigned conversion, ports being declared but not read, asynchronous reset, and write-write race for signals can create a functional but not well-developed code. Warnings are just a preference of what good System Verilog code should be, however, they do not affect the code's ability to run a simulation.

The "near-miss phenomenon" observed with Qwen-Coder models: several designs failed due to minor syntactic or initialization errors (e.g., undeclared wires, inferred latches), indicating partial but incomplete protocol understanding. This suggests potential for "LLM + post-processing" strategies (e.g., automated lint-fix or human-in-the-loop design repair), especially when used with spec conditioning.

Interestingly, varying Clock Polarity and Clock Phase configurations in SPI (while semantically similar) led to structurally different implementations across generations. This reflects a lack of canonical understanding of protocol design patterns and could affect downstream reproducibility.

Finally, the core observation is that while LLMs might be able to generate functional code, they largely struggle with generating synthesizable code. This highlights the value of our staged evaluation approach, where waveform-level simulation acts as the definitive check, beyond syntax and synthesis, to catch timing violations, protocol misbehaviors, and other functional errors that don't surface during earlier stages.

## V. CONCLUSION

We presented the first benchmark designed to evaluate large language models on HDL-based communication protocol generation. Our results show that while larger models

like QwenCoder2.5-32B and GPT-4.1 produce syntactically correct SystemVerilog, they often fail to meet functional and synthesis-level correctness. The introduction of specification-aware prompting improved both the structure and utility of generated designs, particularly for well-documented protocols like SPI and I²C. However, complex protocols such as AXI remained challenging across all models. Our evaluation pipeline spanning linting, synthesis, and waveform simulation seemed to be critical for identifying errors that surface only in post-synthesis or functional testing. These findings highlight the current limitations of LLMs in hardware design and point to the need for more domain-specific tuning and model refinement.

## VI. Limitations and Future Work

Our benchmark focuses on a fixed set of four communication protocols: SPI, I²C, UART, and AXI, which, while representative, do not fully capture the broader spectrum of industrial HDL design, including bus arbiters, memory controllers, and custom pipelines. The current evaluation is limited to module-level designs and does not include integration-level or system-level behavior, such as protocol interoperability or backpressure handling.

Our future work aims to include full power, performance, and area (PPA) analysis or resource usage across different FPGA architectures. Finally, we aim to extend this benchmark with deployment pipelines that test generated designs directly on FPGA hardware and sensor interfaces to evaluate physical correctness in real-world scenarios.

## Acknowledgments

This work was partially supported by the ELLIOT Grant funded by the European Union under grant agreement No. 101214398.

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
