# OpenReview forum: "ProtocolLLM: RTL Benchmark for SystemVerilog Code Generation of Communication Protocols"
_iscaconf.org/ISCA/2025/Workshop/MLArchSys — MLArchSys 2025 Oral_

### Official Review · Reviewer_A4KU · 2025-05-10
**Strong paper which defines a benchmark suite and then shows the results of using this benchmark in 72 tests.**

**Confidence:** 3
**Rating:** 6

**Detailed Feedback And Questions For Authors:**

I'm curious about the many mentions of "(Warn)" - What sort of warnings and how severe.

Perhaps not within the scope of this paper, but I wonder how close we are to having a model that completely passes the benchmark.  And then what other benchmarks might be necessary to advance the state of the art?

**Top Reasons To Accept The Paper:**

The authors recognized a major need, the need to measure performance of LLMs in the generation of HDL, and created a benchmark suite.  They tested four widely used protocols (not niche protocols), and three common models.  In other words, they recognized a problem, defined a benchmark in a very justifiable way, and used common models and protocols to run their benchmarks - 72 different tests.

Seems like a sound methodology addressing a huge need regarding a hot topic, and very relevant to MLArch.

**Top Reasons To Reject The Paper:**

Societal Impact was supposed to be addressed.  Not a show-stopper in my book.

---

### Official Review · Reviewer_Wfyj · 2025-05-16
**ProtocolLLM Review**

**Confidence:** 4
**Rating:** 6

**Detailed Feedback And Questions For Authors:**

This paper presents a benchmark for evaluating the capabilities of LLMs in generating SystemVerilog, building from standard communication protocols. Building a standardized benchmark suite for LLM Verilog generation is an important task, and this paper takes a good first step in formulating this proposal.

More detail on the prompting would be really helpful to better understand the context. The authors mention that "standard prompting" is used, and I think further describing what is in that prompting (or providing an example in the appendix) would really help the reader understand what the LLM is provided. Similarly, references to what the "spec-assisted generation" looks like will also help here. I think it would be helpful to explain how the prompts were constructed, and provide details on the RAG setup used.

The "near-miss phenomenon" is interesting. The authors suggest using "automated lint-fix or human-in-the-loop design repair" in the future, but I wonder if there are a set of common issues that are near-misses and if this could more simply be fixed with augmenting the prompting to cover these common misses.

This is a wording nitpick, but I would not call the last check "Waveform analysis" as that implies that you are  always manually checking the waveforms. In reality, this check is likely done with an automated push-button testbench in Verilog, and you are only checking the waveforms when the results do not match the expectation set in the testbench. I would call this "Functionality check" or something similar instead as first, that gets to the purpose of the check and not how you do the check, and second, I also think it is not entirely accurate to say that how you do the check is always through waveform analysis.

I am curious how good the LLMs are at generating _testbenches_ for these communication protocols as well -- also including testbenches in the benchmark suite could be valuable, since this also evaluates whether the LLM understands enough about the communication protocol and Verilog to build a robust testbench that tests functionality as well as corner cases well.

I would recommend adding a "quality" check to this benchmark suite. Often, generated Verilog is not very easy to read by humans. I realize that this can be a difficult metric to quantitatively evaluate, but I think a qualitative discussion of the quality of the generated Verilog would be valuable to include.

**Top Reasons To Accept The Paper:**

- Building a benchmark suite to evaluate the capabilities of LLMs in generating Verilog is an increasingly important task. The authors build their benchmark based on standard, commonly-used communication protocols, which is a good starting place.
- The three stages in the evaluation -- lint check, synthesis with DC, functional check -- are a strength of the paper. Particularly, synthesis is an important step for generating Verilog that can actually be used in downstream physical design flows.

**Top Reasons To Reject The Paper:**

- Not much detail is provided on the prompt engineering required, so it is difficult for the reader to understand the context provided to the LLM before it attempts to generate Verilog.

---

### Official Review · Reviewer_5kaf · 2025-05-18
**Nice work --- would like more detail**

**Confidence:** 4
**Rating:** 6

**Detailed Feedback And Questions For Authors:**

I really like this---I think it's a good workshop paper, and has the beginnings of a very strong full paper.

A few questions and suggestions:
1. Your benchmark doesn't have a numeric score. I could score it numerically by counting the checks, but that gives a very quantized score, more like a pass/fail. This exposes the risk of false ties, where two candidates perform differently but within the quantization level of the benchmark (pass/fail on broader tests). It would be great to have numeric scores, like "how many lint errors were generated" or "how many test cases failed.
2. How stringent are the tests? If the tests are automated, are you checking non-fault cases only or trying a wider range of cases?
3. You mention a temperature of zero for reproducibility. If cost permits, would it be possible to additionally quantify the run-to-run variation from a non-zero temperature? Will zero temperature give worse outputs than a higher temperature?
4. Your protocol descriptions don't describe their functionality well, and a table of core specifications could be helpful here. For example, it seems that you are varying some non-trivial SPI parameters, but it's unclear from the paper what you're doing for UART. UART can be very simple (fixed baud, fixed framing, no flow control) to complicated (baud detection, parity, buffering, flow control).
5. What is being generated? What is the ratio of prompt to generated verilog? It would be great to have a table listing the number of lines of code generated and LUTs/gates/FLOPs. This will help situate the work (along with the previous point) so that someone who is experienced with RTL but not with LLMs can understand the capabilities of these models and how much they're actually generating.
6. Do you evaluate protocol performance at all? I know that you're evaluating functionality, which is obviously the first step, and that much of what you're generating now is fairly simple from a performance perspective. However, it would be interesting to see how fast the designs run (cycle time and number of cycles), and if there's any variation across models.

Textual nits:

p2 "master controllers in embedded [systems]"

p3 "on a 3 three-prong basis"

p3 "testbenches the randomly generate"

p4 spell out CPOL/CPHA

**Top Reasons To Accept The Paper:**

This is an interesting paper on a highly relevant area. Automatic generation of HDL has a lot of industrial potential, and the paper provides some interesting evaluation of leading LLMs.

**Top Reasons To Reject The Paper:**

The benchmark suite could be improved to increase it's specificity and scoring range.

---

### Official Review · Reviewer_yNKk · 2025-05-19
**ProtocolLLM: Benchmark for SystemVerilog Code Generation for LLMs**

**Confidence:** 3
**Rating:** 5

**Detailed Feedback And Questions For Authors:**

Thanks for the submission.

Great initiative!

My main concern is the evaluation part - it would be good to address the issue or mention the intended path for this benchmark.

I am not sure what the copyright requirement of the AXI spec would be. Considering that it is a copyright of a company, you may not be able to release it along with the benchmark.

How does this differ from VerilogEval apart from the evaluation aspect?

**Top Reasons To Accept The Paper:**

1. The paper introduces a benchmark for LLMs specifically for SystemVerilog - earlier works may have mostly focused only on Verilog.
2. This work also appears to introduce more rigorous protocol-specific testing with a three-stage evaluation pipeline.
3. The authors provide the system prompt with example results from the three models they evaluate.

**Top Reasons To Reject The Paper:**

1. I am mainly concerned about the evaluation. Considering that the benchmark relies on actual EDA tools and waveform analysis - how will benchmark tester do that without applying for access to such EDA tools and expertise of waveform evaluation. If this a benchmark that can explicitly be tested by the authors only then they should probably keep the benchmark hidden or maybe some canary strings.
2. It would have been beneficial to assign a score to each model rather than a simple YES/NO evaluation. Some ROUGE/F1 score for the code generated compared to golden outputs would be better.